# Changes in Relationship between Forest Biomass Productivity and Biodiversity of Different Type Subtropical Forests in Southern China

Wei Xu, Ping Zhou *, Miguel Ángel González-Rodríguez, Zhaowei Tan, Zehua Li and Ping Yan

Guangzhou Institute of Geography, Guangdong Academy of Sciences, Guangzhou 510070, China; xuwei12@gdas.ac.cn (W.X.); miguel@gdas.ac.cn (M.Á.G.-R.); zhaoweit@gdas.ac.cn (Z.T.); zehual@gdas.ac.cn (Z.L.); yanping@gdas.ac.cn (P.Y.)
* Correspondence: pzhou@gdas.ac.cn

**Abstract:** Forest productivity is influenced by various factors, including biodiversity, environmental factors, functional traits, and forest types. However, the relative importance of these factors in determining the productivity of subtropical forests in southern China remains controversial. In this study, we analyzed a dataset of 24 forest plots from four subtropical forest types in the Nanling Mountains with the main goal of identifying and quantifying the relative contribution of the main driving factors of forest productivity in these forests. Generalized linear regression and structural equation modeling were used to examine the relationship between forest biomass productivity (aboveground, belowground and total), biodiversity (taxonomic diversity, phylogenetic diversity and functional diversity), and environmental variables (i.e., physiography and climate). The results indicated that both environmental factors and biodiversity played pivotal roles in explaining the biomass productivity of the Nanling subtropical forests. Environmental factors had the greatest influence on total productivity, while the impacts of different types of biodiversity on various productivity components (aboveground and belowground) varied notably. Taxonomic diversity showed the strongest positive effect on the aboveground and belowground biomass productivity. However, phylogenetic and functional diversity had negative effects on productivity. Furthermore, these relationships also exhibited variations when considering different altitude gradients, with low altitudes generally leading to negative biodiversity–productivity correlations. We contextualized our results regarding the three state-of-the-art theories about biodiversity–productivity relationships (selection probability, niche complementarity, and biomass ratio) and concluded that both selection probability and niche complementarity are the driving mechanisms of productivity in the subtropical forests of the Nanling Mountains. This study offers valuable insights into the functioning and biodiversity mechanisms of subtropical forest ecosystems in southern China.

**Keywords:** biodiversity; subtropical forests; Nanling Mountains; biomass productivity





## 1. Introduction

The relationship between forest biodiversity and ecosystem functions represents a critical and extensively investigated aspect of forest ecosystems [1,2]. With the increasing impacts of global climate change and human activities, this relationship has attracted marked attention [3–5]. Numerous studies have shown that biodiversity plays a crucial role in forest productivity, as forests with high biodiversity tend to have higher productivity and be better providers of ecosystem services [6–8]. However, the relationship between biodiversity and productivity is not always straightforward, as research indicates that the correlation may not be linear [9,10]. There are instances where high biodiversity does not necessarily lead to a proportional increase in productivity [11]. In fact, some studies indicate that productivity may decrease after surpassing a specific biodiversity

threshold [12,13]. Therefore, it is essential to further explore the complex relationship between forest biodiversity and productivity [9,14].

Many studies have tackled the complex dynamics linking productivity and biodiversity [6,15]. The main hypotheses for how biodiversity affects forest productivity are the niche complementary effect, the selection probability effect, and the biomass-ratio effect. The niche complementary effect hypothesizes that high biodiversity can increase forest productivity through niche differentiation and promotion [11]. The selection probability effect suggests that higher species richness boosts community productivity by raising the likelihood of having high-yielding species [16]. Under this situation, the impact of functional diversity on productivity is notably weaker than that of taxonomic diversity. The biomass-ratio hypothesis, on the other hand, postulates that the characteristics of the dominant species within the community drive ecosystem functions, which is a mechanism that does not conflict with niche complementarity [17]. These three ecosystem functioning hypotheses could coexist to some extent [18,19], though their relative significance remains an ongoing discussion [17,20].

Further investigation into the association between biodiversity and productivity requires a deeper understanding of the role environmental factors play [2,21,22]. These factors, such as climate, soil, and physiography, have a strong influence on the relationship between biodiversity and productivity in forest ecosystems [2]. Climatic and soil factors [23,24], as well as light and moisture conditions [25,26], are examples of environmental factors that can have a significant impact on the distribution, growth, and competition of forest ecosystems [19,27], which in turn affects the interaction between forest biodiversity and productivity [15,24]. Numerous studies have revealed that the relationship between biodiversity and productivity varies under different environmental conditions [21,27,28]. For example, as altitude increases, forest ecosystems show significant changes in climate, soil, and vegetation [29], and these changes ultimately affect forest biodiversity and productivity [30]. In addition, several studies have shown that the relationship between forest biodiversity and productivity is significantly impacted by changes in altitude [31,32]. In these studies, the relationship between biodiversity and productivity displayed negative or unimodal patterns in response to increasing altitude [31,33,34]. Changes in environmental conditions, such as variations in temperature and precipitation that covariate with altitude, might also affect forest biodiversity and productivity [19,33,35]. Moreover, altitude changes can also have an impact on forest vegetation types, species composition, and ecosystem functions [36]. Consequently, exploring the relationship between biodiversity and productivity along altitude gradients might be essential to understand ecological processes and response mechanisms in forest ecosystems [33,36].

In the current study, our objective was to examine the impact of biodiversity and environmental factors on the productivity of various subtropical forest communities in the Nanling Mountains of southern China. Based on the monitoring data of 24 fixed plots of four different forest types in the Nanling Mountains, we selected a number of biodiversity indices, including taxonomic, phylogenetic, and functional diversity. Forest productivity was estimated in terms of aboveground, belowground, and total tree biomass. The influence of biodiversity indices and environmental factors on forest productivity was analyzed through generalized linear regression and structural equation models to examine three main questions: (1) the impact of various biodiversity indices on forest productivity, (2) the relative contribution of environmental factors and biodiversity to biomass productivity of forest communities, and (3) the role of altitude in shaping the biodiversity–productivity relationship.

## 2. Materials and Methods

### 2.1. Study Area

The study area is located in the Nanling National Nature Reserve, in the middle of the Nanling Mountains, in the north of Guangdong Province, southern China. It spans three counties, namely Ruyuan, Yangshan and Lianzhou, with geographical coordinates

24°37′ N–24°57′ N, 112°30′ E–113°04′ E, and covers an area of 58,400 hectares. The highest peak, Shikeng Kong, is 1902 m above sea level and has a relative elevation difference of 1489 m. The study site was established in 2017 following the CTFS (Center for Tropical Forest Science of The Smithsonian Tropical Research Institute) standard census methods, and vegetation types include valley evergreen broad-leaved forest (VBF), mountain evergreen broad-leaved forest (MBF), evergreen coniferous broad-leaved mixed forest (EMF), and evergreen broad-leaved dwarf forest (DWF). The study region exhibits specific monsoon climate between Central Asia and South Asia. The annual average temperature is 17.7 °C, with annual minimum and maximum temperatures of −4.2 °C and 34.4 °C, and an annual average frost-free period of 276 days. The annual average precipitation is 1705 mm, with the annual average relative humidity of 84%. Most of the annual rainfall is distributed between March and October, making up roughly 82% of the total annual rainfall. The soils of the reserve region primarily include red soil, yellow soil, and mountain scrubby-meadow soil [37].

### 2.2. Data Collection

Data were collected from 24 permanent plots of 40 m × 40 m in the Nanling National Nature Reserve (Figure 1), which were surveyed in the years 2017 and 2020. Six VBF plots, nine MBF plots, six EMF plots, and three DWF plots were among the various vegetation types found in the plots. Each fixed plot was divided into four 20 m × 20 m quadrats as vegetation survey units, except for the mountain scrubby-meadow fixed plot, which was divided into four 10 m × 10 m quadrats. The four corners of the quadrats were marked and fixed with cement piles. The geographic location (latitude, longitude) and topographic variables (altitude, slope, aspect) were collected for each plot (Supplementary Table S1). All woody stems with ≥1 cm diameter at a breast height of 1.3 m (DBH) in the plot were mapped, identified, and their DBH, tree height, and crown width were measured.

### 2.3. Productivity Calculations

The woody biomass of all individual trees was calculated using species-specific allometric growth equations [38–41]. All tree individuals with a DBH ≥5 cm were used in the analysis. For each tree, the variation in tree biomass over a three-year period was evaluated to determine the aboveground biomass productivity (ABP), belowground biomass productivity (BBP), and total biomass productivity (TP). A total of 18,715 tree individuals, belonging to 201 species, were counted throughout both inventories, excluding dead trees and recruits, which were not taken into account. In the end, the biomass increments from 2017 to 2020 were used to calculate each plot's forest biomass productivity. Additional comprehensive details regarding the productivity estimation can be found in the Supplementary Table S2.

### 2.4. Biodiversity Indices and Functional Traits

Three different criteria (taxonomic species diversity, phylogenetic diversity and functional diversity) were used to examine the effect of biodiversity on forest productivity. Species richness (number of species per plot, S), the Shannon–Weaver diversity index (H), the Simpson diversity index (DS), and Pielou evenness index (J) were used to quantify taxonomic species diversity (TD).

For phylogenetic diversity (PD), we used the software of Phylomatic-awk-1.0.0 (http://www.phylodiversity.net accessed on 1 June 2023) [42]. The tree with branch lengths generated by Zanne et al. [43], which combined the DNA marker data with a molecular phylogeny for land plants, was more accurate than using the tree APG III system as a skeleton, and was consequently selected for estimating diversity indices. Three phylogenetic diversity indices were calculated based on this phylogenetic tree (Figure S1) as follows: the phylogenetic diversity (pd), defined as the total branch length across the phylogenetic tree of all the species within the community [44]; the mean pairwise phylogenetic distance (mpd) between all the species in each community; and the mean nearest taxon distance

(mntd), defined as the mean distance separating each of the species within the community from its closest relative [43]. The phylogenetic analysis was implemented in Phylocom version 4.2 [42] and the R package "picante" [45].

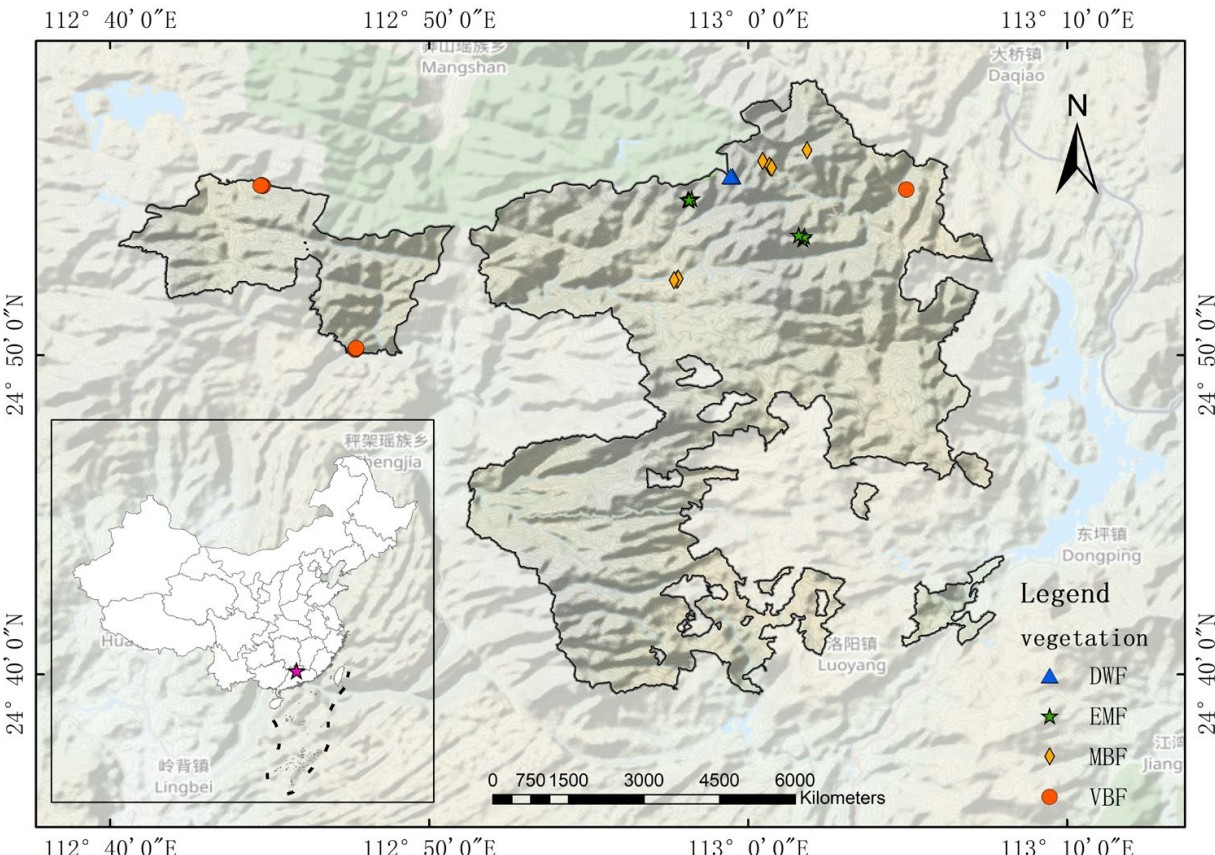

**Figure 1.** Spatial distribution of the forest plots in the Nanling National Nature Reserve, southern China. Abbreviations: DWF, evergreen broad-leaved dwarf forest; EMF, evergreen coniferous broad-leaved mixed forest; MBF, mountain evergreen broad-leaved forest; VBF, valley evergreen broad-leaved forest.

Functional diversity indices were estimated using the functional traits which better reflected the growth capacity of the considered species. The functional traits of 201 tree species were utilized, including total carbon content, total nitrogen content, total phosphorus content, total potassium content, sulphur content, calcium content, magnesium content, gross caloric value, ash content of steam, leaves, and roots (Table 1). For the 56 species for which trait measurement data were available, we used the means of genera to approximate the traits of the 155 unmeasured species. We applied the means of total data to unmeasured tree species that did not belong to any measured genus.

Using the dbFD function of the R package 'FD', three functional diversity indices (FD) were used: functional evenness (FEve) and functional divergence (FDiv) proposed by Villeger et al. [46], and functional dispersion (FDis) proposed by Laliberté and Legendre [47]. Community-weighted means (CWM) of each trait, weighted by the density of each species for each plot, were also calculated using the R package "FD" [48].

**Table 1.** Summary of functional traits data from the 24 plots in Nanling's subtropical forests.

| Variables | Abbreviation | Units | Ecological Roles | $\bar{x} \pm$ **std** |
|---|---|---|---|---|
| Stem Total Carbon content | stc | % | stem structure and function | 50.76 ± 0.78 |
| Stem Total Nitrogen content | stn | g/kg | plant growth and photosynthesis | 12.81 ± 0.62 |
| Stem Total Phosphorus content | stp | g/kg | reproduction and energy transfer | 1.25 ± 0.06 |
| Stem Total Potassium content | stk | k/kg | enzyme activation and the transport of nutrients and water | 7.08 ± 0.49 |
| Stem Sulphur content | ssc | g/kg | component of proteins | 1.79 ± 0.4 |
| Stem Calcium content | scc | g/kg | prevention of plant diseases and maintenance of cell structure | 15.63 ± 3.22 |
| Stem Magnesium content | smc | mg/kg | photosynthesis | 2.16 ± 0.28 |
| Stem Gross caloric value | sgc | KJ/g | energy content | 26.28 ± 1.88 |
| Stem Ash content | sac | % | inorganic mineral content | 7.33 ± 0.15 |
| Leaves Total Carbon content | ltc | % | leaves structure and photosynthesis | 50.76 ± 0.78 |
| Leaves Total Nitrogen content | ltn | g/kg | key component of chlorophyll, photosynthesis | 14.03 ± 0.6 |
| Leaves Total Phosphorus content | ltp | g/kg | energy transfer and metabolic processes | 1.37 ± 0.07 |
| Leaves Total Potassium content | ltk | k/kg | osmoregulation and stomatal control | 8.34 ± 0.75 |
| Leaves Sulphur content | lsc | g/kg | defense mechanisms | 2.25 ± 0.55 |
| Leaves Calcium content | lcc | g/kg | resistance | 19.82 ± 3.75 |
| Leaves Magnesium content | lmc | mg/kg | key component of chlorophyll, photosynthesis | 2.56 ± 0.3 |
| Leaves Gross caloric value | lgc | KJ/g | energy content | 19.29 ± 2.19 |
| Leaves Ash content | lac | % | inorganic mineral content | 8.22 ± 0.31 |
| Roots Total Carbon content | rtc | % | energy storage and resource acquisition of roots | 50.67 ± 0.88 |
| Roots Total Nitrogen content | rtn | g/kg | nutrient uptake and plant growth | 11.89 ± 0.67 |
| Roots Total Phosphorus content | rtp | g/kg | energy transfer, nutrient uptake, and root growth | 1.14 ± 0.06 |
| Roots Total Potassium content | rtk | k/kg | water and nutrient uptake, osmoregulation | 6.33 ± 0.49 |
| Roots Sulphur content | rsc | g/kg | resistance and mycorrhiza fungi | 1.48 ± 0.38 |
| Roots Calcium content | rcc | g/kg | root development | 14.23 ± 3.27 |
| Roots Magnesium content | rmc | mg/kg | metabolic processes | 1.88 ± 0.33 |
| Roots Gross caloric value | rgc | KJ/g | energy content | 21.1 ± 2.04 |
| Roots Ash content | rac | % | inorganic mineral content | 6.68 ± 0.17 |

*2.5. Environmental Variables*

The physiographic conditions and meteorological information were included in the environmental variables. Physiographic conditions comprised altitude, slope, south–north aspect (sn_aspect), west–east aspect (we_aspect), the Topographic Position Index (TPI), and the Terrain Ruggedness Index (TRI), derived from the 250 m SRTM (Shuttle Radar Topography Mission) Digital Elevation Model for China [49], and computed using the R language package "raster" [50]. Moreover, we extracted climatic variables for the 2017–2020 period, including mean annual precipitation (mnp), mean annual temperature (mnt), January's minimum temperature (min_JAN), and July's maximum temperature (max_JU) from the Monthly Spatial Interpolation Dataset of Meteorological Elements in China [51]. In this study, the altitude ranges of 400 to 850 m, 850 to 1400 m, and 1400 to 1700 m were categorized as low-altitude, mid-altitude, and high-altitude ranges, respectively. The 7 plots in low-altitude gradient belong to subtropical VBF(6) and MBF(1), since VBF can still keep the forest warm and humid in the dry season, and the vegetation

shows some rainforest characteristics, such as the plate root phenomenon, and the trees are mainly composed of families Lauraceae, Duyingaceae, and Myrtaceae. The mid-altitude gradient has 11 plots, categorized as MBF(7) and EMF(4). MBF is mainly distributed in low hills below 1400 m in Nanling Mountains, where it is occasionally accompanied by deciduous tree species, and most of them belong to the family Fagaceae. And EMF is mainly distributed on steep slopes at an altitude of 1000–1500 m, and the common species are mainly *Pinus kwangtungensis*, *Chamaecyparis hodginsii*, and *Nothotsuga longibracteata*. Within the high-altitude gradient, six plots are designated under DWF(3), EMF(2), and MBF(1). DWF is primarily distributed at mountains exceeding 1500 m in altitude, where the weather is permanently windy and cloudy. Trees in this area are small, usually between 4 and 8 m tall, and the predominant species are rhododendrons (*Rhododendron* sp.).

### 2.6. Statistical Analysis

A logarithmic transformation was applied to the forest productivity data prior to data analysis. In addition, all environmental variables, diversity indices, community-weighted trait values, and data of forest productivity were standardized. Pearson correlation coefficients of environmental and biodiversity indices can be found in the Supplementary Table S3. Multicollinearity between variables was evaluated based on the Variance Inflation Factor (VIF), with a threshold set at 3, using the "CAR" R package [52]. Following this preprocessing process, the final groups of variables were identified: environmental variables (ENV, e.g., mnp, min_JAN, slope, aspect, TPI, and TRI), taxonomic species diversity (TD, e.g., DS, J, and S), phylogenetic diversity (PD, e.g., pd, mpd, and mntd), functional diversity (FD, e.g., FEve, FDiv, and FDis), CWM of specific traits (as CWM.stc, CWM.stk, CWM.sm, CWM.sgc, CWM.ltc, CWM.ltn, CWM.lgc, CWM.rtn, CWM.rtp, CWM.rtk, CWM.rgc), and forest biomass productivity data (ABP, BBP, TP). The effects of each variable on forest biomass productivity were investigated using general linear regression analysis. The "glmm.hp" R package was used to rank the explanatory power of various groups on productivity [53].

A structural equation modeling (SEM) approach was used to analyze how environmental and biodiversity variables affected ABP, BBP, and TP. We set up two theoretical models that differ in the order in which productivity affects the aboveground and belowground components. Model *A* assumes that the aboveground productivity affects the belowground productivity first, while model *B* assumes that the belowground productivity influences the aboveground productivity first. The structural equation modeling was implemented using the "lavvan" R package [54]. The statistical analyses were all carried out using R 4.2.3 [55].

## 3. Results

### 3.1. Relationship between Predictive Variables and Forest Productivity

When examining the relationships between functional aspects and forest productivity, marked differences were found in the associations with aboveground and belowground productivity (Table 2). Specifically, CWM.rtn explained the majority of the variance in ABP (coefficient = 0.14, $R^2$ = 0.31, $p < 0.01$), while FD exhibited a significant negative correlation with ABP (coefficient = $-0.13$, $R^2$ = 0.26, $p < 0.05$). Notably, a negative relationship between BBP and mntd (coefficient = $-0.17$, $R^2$ = 0.45, $p < 0.001$) was found, but positive correlations were found between BBP and CWM.ltn and CWM.rtn (coefficients: $-0.12$, $R^2$ = 0.23, $p < 0.05$; $-0.15$, $R^2$ = 0.38, $p < 0.001$). Only TPI displayed a statistically significant positive correlation with TP (coefficient = 0.25, $R^2$ = 0.99, $p < 0.01$).

Regarding the analysis of effects on productivity based on grouped multiple linear regression, environmental factors were the best explainers of TP (51.52%, $p = 0.08$), while biodiversity variables demonstrated significant explanatory power over both ABP and BBP. Interestingly, the highest explanatory power for ABP (45.01%, $p < 0.05$) and BBP (79.3%, $p < 0.001$) was found to be attributed to taxonomic species diversity. Additionally, compared to separate ABP and BBP, the CWM values of individual functional traits had a slightly higher explanatory power for TP (Table 3).

**Table 2.** Bivariate relationships between productivity (log-transformed) and environment factors, biodiversity, and community-weighted means of each trait.

| Variables | | ABP | | | BBP | | | TP | | |
|---|---|---|---|---|---|---|---|---|---|---|
| | | Coefficient | $R^2$ | *p*-Val. | Coefficient | $R^2$ | *p*-Val. | Coefficient | $R^2$ | *p*-Val. |
| ENV | mnp | 0.29 | 0.08 | 0.16 | 0.2 | 0.04 | 0.33 | 0.29 | 0.09 | 0.17 |
| | min_JAN | 0.27 | 0.07 | 0.21 | 0.18 | 0.03 | 0.41 | 0.26 | 0.07 | 0.21 |
| | slope | 0.11 | 0.01 | 0.6 | 0.07 | 0.005 | 0.75 | 0.11 | 0.01 | 0.6 |
| | sn_aspect | −0.11 | 0.01 | 0.63 | −0.08 | 0.007 | 0.7 | −0.11 | 0.01 | 0.62 |
| | we_aspect | −0.21 | 0.04 | 0.33 | −0.23 | 0.05 | 0.27 | −0.22 | 0.04 | 0.31 |
| | TPI | −0.1 | 0.1 | 0.63 | −0.13 | 0.02 | 0.54 | −0.11 | 0.01 | 0.61 |
| | TRI | 0.11 | 0.01 | 0.61 | 0.21 | 0.05 | 0.32 | 0.13 | 0.02 | 0.55 |
| TD | DS | 0.29 | 0.08 | 0.17 | 0.54 | 0.29 | 0.01 * | 0.34 | 0.11 | 0.11 |
| | J | 0.28 | 0.08 | 0.18 | 0.56 | 0.31 | 0.004 | 0.33 | 0.11 | 0.11 |
| | S | 0.21 | 0.04 | 0.33 | 0.41 | 0.17 | 0.05 | 0.24 | 0.06 | 0.25 |
| PD | pd | 0.01 | 0.001 | 0.9 | 0.17 | 0.03 | 0.43 | 0.14 | 0.01 | 0.9 |
| | mpd | −0.26 | 0.07 | 0.21 | −0.14 | 0.02 | 0.5 | −0.26 | 0.07 | 0.22 |
| | mntd | −0.32 | 0.1 | 0.13 | −0.63 | 0.39 | 0.001 ** | −0.37 | 0.14 | 0.07 |
| FD | FEve | −0.11 | 0.01 | 0.61 | 0.04 | 0.001 | 0.85 | 0.09 | 0.001 | 0.66 |
| | FDiv | −0.002 | 0.001 | 0.9 | −0.23 | 0.06 | 0.26 | −0.03 | 0.001 | 0.89 |
| | FDis | −0.45 | 0.21 | 0.03 * | −0.63 | 0.4 | <0.001 *** | −0.5 | 0.25 | 0.01 * |
| CWM | CWM.stn | 0.42 | 0.18 | 0.04 * | 0.67 | 0.45 | 0.001 ** | 0.47 | 0.22 | 0.02 * |
| | CWM.stp | −0.003 | 0.001 | 0.9 | −0.27 | 0.08 | 0.2 | −0.04 | 0.002 | 0.9 |
| | CWM.stk | 0.1 | 0.01 | 0.65 | 0.31 | 0.1 | 0.15 | 0.13 | 0.02 | 0.54 |
| | CWM.sgc | 0.04 | 0.001 | 0.9 | −0.01 | 0.001 | 0.9 | 0.03 | 0.001 | 0.9 |
| | CWM.ltc | −0.04 | 0.002 | 0.84 | −0.22 | 0.05 | 0.29 | −0.07 | 0.01 | 0.74 |
| | CWM.ltn | 0.28 | 0.08 | 0.17 | 0.51 | 0.26 | 0.01 * | 0.33 | 0.11 | 0.11 |
| | CWM.rtc | −0.16 | 0.02 | 0.46 | −0.27 | 0.07 | 0.2 | −0.18 | 0.03 | 0.4 |
| | CWM.rtn | 0.5 | 0.25 | 0.01 * | 0.63 | 0.4 | <0.001 *** | 0.54 | 0.29 | 0.01 * |
| | CWM.rtp | 0.17 | 0.03 | 0.43 | −0.02 | 0.001 | 0.9 | 0.15 | 0.02 | 0.5 |
| | CWM.rgc | 0.17 | 0.03 | 0.42 | 0.18 | 0.03 | 0.4 | 0.18 | 0.03 | 0.4 |

\* indicates *p* < 0.05, \*\* indicates *p* < 0.01, \*\*\* indicates *p* < 0.001.

**Table 3.** Summary of regression model for the effects of environment factors, taxonomic diversity, phylogenetic diversity, functional diversity, and community-weighted means of each trait on productivity in Nanling Mountain forests.

| Variable | VIF | Coefficient | ABP Individual Effect (%) | *p*-Value | Coefficient | BBP Individual Effect (%) | *p*-Value | Coefficient | TP Individual Effect (%) | *p*-Value |
|---|---|---|---|---|---|---|---|---|---|---|
| ENV | 1.31 | 0.002 | 0.18 | 0.90 | −0.01 | 2.59 | 0.48 | 0.03 | 51.52 | 0.08 |
| TD | 2.27 | 0.06 | 45.01 | 0.06 | 0.09 | 79.3 | 0.001 *** | 0.03 | 11.43 | 0.40 |
| PD | 1.28 | −0.09 | 26.8 | 0.04 * | −0.14 | 26.1 | 0.001 *** | −0.05 | 9.27 | 0.30 |
| FD | 1.32 | −0.11 | 18.73 | 0.04 * | −0.11 | 0.81 | 0.01 ** | 0.003 | 7.39 | 0.96 |
| CWM | 2.44 | −0.002 | 0.35 | 0.86 | −0.01 | 1.43 | 0.60 | 0.006 | 2 | 0.72 |
| R2 | | | 0.27 | | | 0.46 | | | 0.22 | |

\* indicates *p* < 0.05, \*\* indicates *p* < 0.01, \*\*\* indicates *p* < 0.001.

### 3.2. The Relative Importance of Environmental Factors and Diversity in Determining Forest Community Productivity

All the predictor variables, including ENV and biodiversity, collectively explained 22% of TP, 58% of ABP, and 52% of BBP, according to the structural equation model A (Figure 2a). TD exhibited a significant positive effect on BBP, while both PD and FD exhibited significant negative effects on BBP. The BBP had a noteworthy positive impact on ABP. Furthermore, TP was significantly positively impacted by ENV.

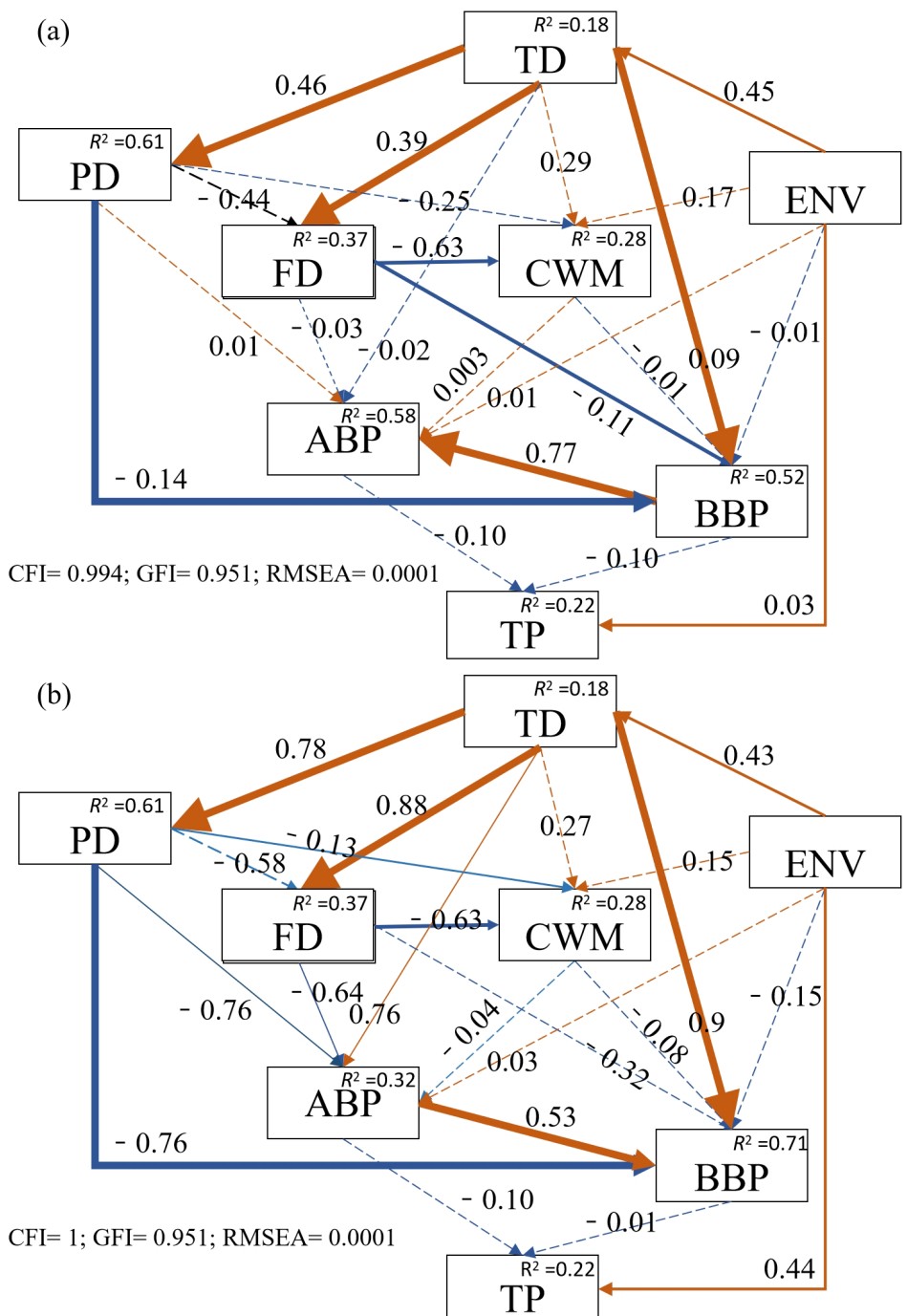

**Figure 2.** Structural equation models that link ENV, TD, PD, FD, and CWM to ABP, BBP, and TP. The coefficients are standardized prediction coefficients for each causal path. (**a**) theoretical model *A*; (**b**) theoretical model *B*. Solid lines represent significant paths ($p \leq 0.05$) and dashed lines indicate non-significant paths ($p > 0.05$). Line color denotes the positive (red) or negative (blue) coefficient values. The thickness of the solid arrows reflects the significance level of the standardized prediction coefficients. $R^2$ represents the proportion of variance explained. Abbreviations: TD, taxonomic species diversity; PD, phylogenetic diversity; FD, functional diversity; ENV, environmental factors; CWM, community-weighted means of specific traits; ABP, aboveground biomass productivity; BBP, belowground biomass productivity; TP, total biomass productivity; CFI, comparative fit index; GFI, goodness-of-fit index; RMSEA, root mean square error of approximation.

For structural equation model B, 22% of TP, 32% of ABP, and 71% of BBP were explained (Figure 2b). Notably, PD showed a significant negative effect on ABP in this

alternative model, and its negative effect on CWM became significant. FD displayed a significant negative effect on ABP, but its negative effect on BBP was not significant. TD exhibited a significant positive effect on ABP.

*3.3. Relationship between Diversity and Forest Productivity across Altitude and Vegetation Classes*

Variations in woody plant productivity were noted in the aboveground, belowground and total productivity across altitude gradients and vegetation types (Figure 3). Specifically, ABP and TP were highest in the low-elevation areas and lowest in the high-elevation sites. Conversely, BBP in mid-elevation forests exceeded that in both low- and high-elevation areas. When comparing productivity among different forest types, we found that MBF exhibited the highest ABP, BBP, and TP, followed by VBF, while tree productivity in DMF was the lowest.

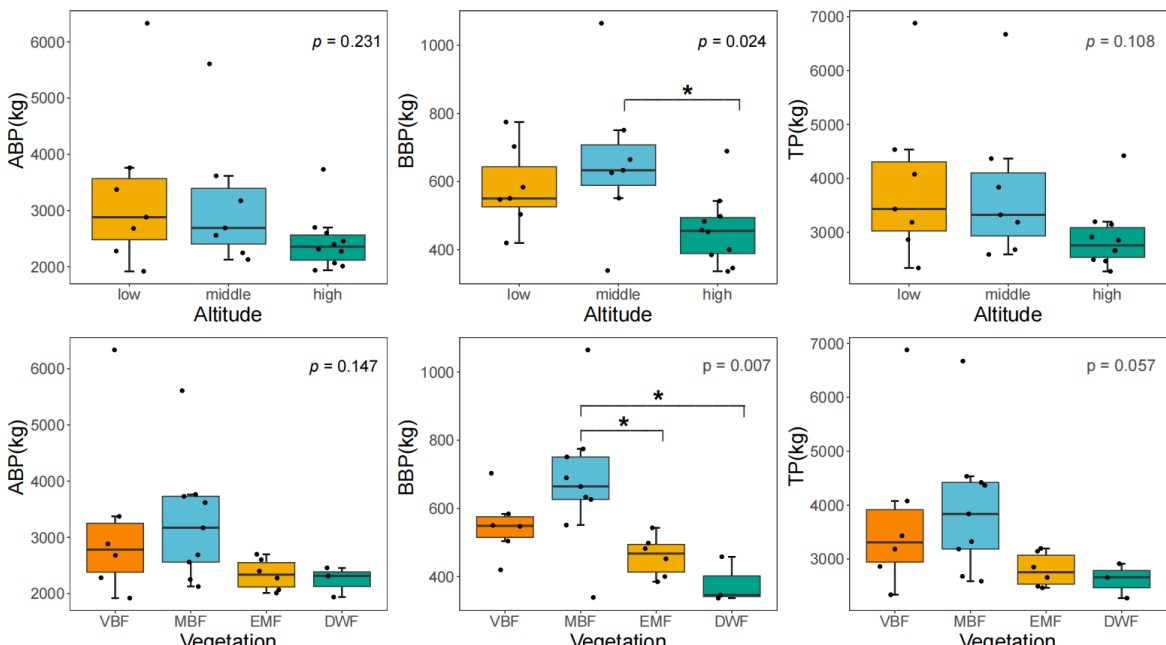

**Figure 3.** Variations in aboveground, belowground, and total productivity across diverse altitudes and vegetation types. Abbreviations: ABP, aboveground biomass productivity; BBP, belowground biomass productivity; TP, total biomass productivity; VBF, valley evergreen broad-leaved forest; MBF, mountain evergreen broad-leaved forest; MBF, mountain evergreen broad-leaved forest; EMF, evergreen coniferous broad-leaved mixed forest; DWF, evergreen broad-leaved dwarf forest. * indicates the significance of difference between the different altitudes or vegetation types ($p < 0.05$).

Vegetation types and altitude classes had varied effects on the relationship between species richness and productivity (Figure 4). In particular, there is a negative correlation at low altitude and a positive correlation at mid-to-high altitude gradients between ABP and species richness. Along the three altitude classes, BBP and species richness have a positive correlation. At low altitude, there is a similar negative association between TP and species richness, but at both mid and high altitude, there are positive correlations. There is a negative association found in VBF between ABP, TP, and species richness, contrary to the other forest community types. In contrast, productivity and species richness show a positive association in MBF, EMF, and DMF.

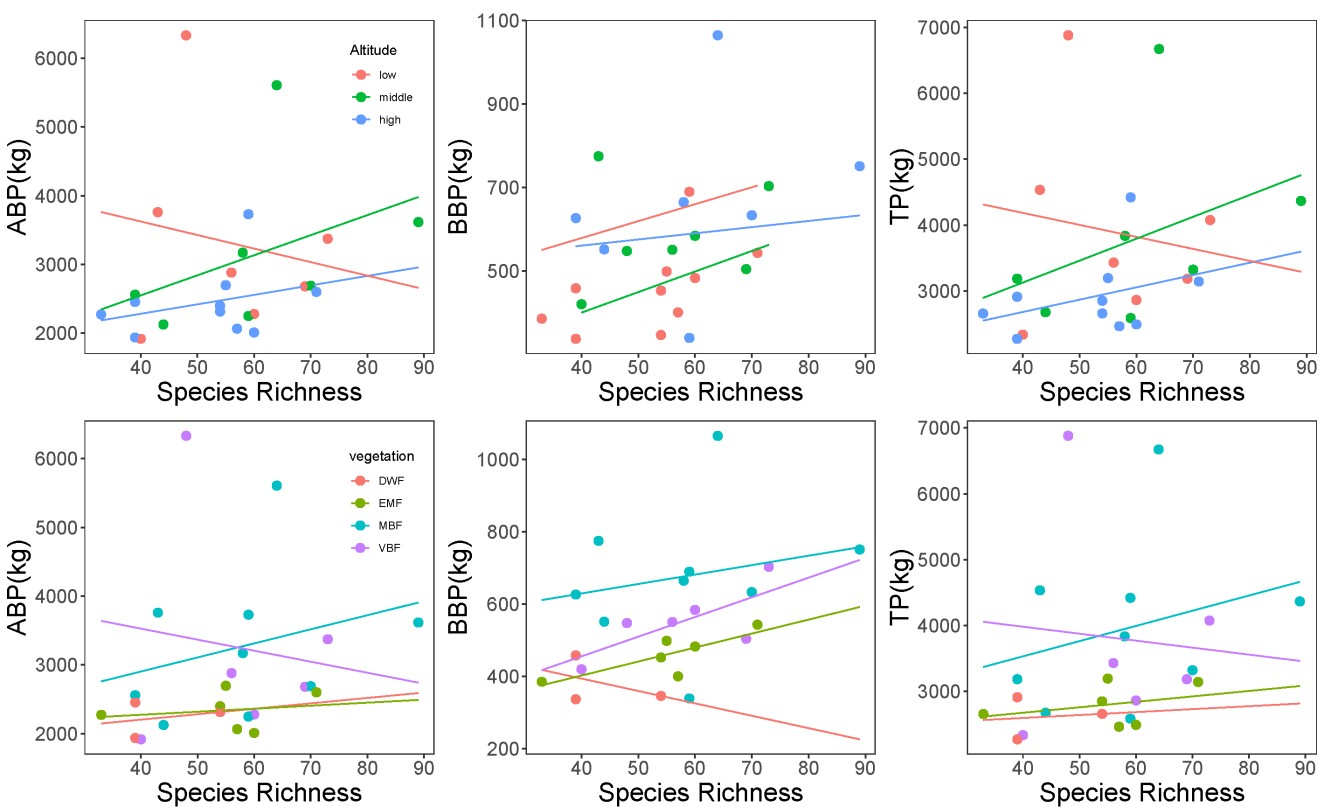

**Figure 4.** Relationship between species richness and aboveground, belowground, and total productivity across diverse altitudes and vegetation types. Abbreviations: ABP, aboveground biomass productivity; BBP, belowground biomass productivity; TP, total biomass productivity; VBF, valley evergreen broad-leaved forest; MBF, mountain evergreen broad-leaved forest; MBF, mountain evergreen broad-leaved forest; EMF, evergreen coniferous broad-leaved mixed forest; DWF, evergreen broad-leaved dwarf forest.

## 4. Discussion

### 4.1. The Effects of Biodiversity on Forest Productivity in Nanling Mountains

In the current study, we estimated the biomass productivity of Nanling Mountains subtropical forests and analyzed its relationship with several taxonomic, phylogenetic, and functional diversity metrics, as well as with a set of functional traits metrics and environmental factors, such as altitude. Overall, our results showed that productivity increases with higher species richness in the subtropical forests of Nanling Mountains, despite the influence of environmental factors and forest types on forest productivity. This is consistent with previous studies finding that forest diversity and biomass productivity have positive a relationship [7,19,56]. Previous research also suggests that the relationship between biodiversity and productivity varied with the spatial scale of investigation [19,57,58]. Our results, showing a positive richness–productivity relationship, were observed at a plot scale of 0.16 hectares, which is consistent with the research of Chisholm et al. [59]. At small scales (e.g., 0.04 hectares), biodiversity and productivity exhibit a positive relationship, but neutral or negative relationships become more common at larger scales (e.g., 0.25 or 1 hectare) [19,59]. Nevertheless, figuring out the specific scale at which biodiversity and productivity become less correlated is a worthy aim to pursue in the future [56].

TD indices displayed positive correlations with biomass productivity. As biodiversity increases, the possibility of complementarity in resource utilization and ecological functions among species rises [60]. Furthermore, the probability of the presence of high-productivity species also increases, contributing to the overall productivity through selection and complementarity effects [61]. However, there are specific phylogenetic diversity and functional diversity indices that exhibit negative relationships with productivity. Specifically, mpd and

mntd show negative correlations with both ABP and BBP. FDis shows significant negative correlations with both ABP and BBP. These patterns suggest significant evolutionary and trait differences between species, which lead to functional overlapping or redundancy and have negative impacts on productivity [34]. It is suggested that dominant functional groups exist within the Nanling forest community, with species clustering towards these groups contributing to higher productivity. Communities with greater evolutionary and functional trait dissimilarity may have negative effects on productivity, that can be explained by the Biomass-Ratio hypothesis [62]. Environmental factors exert the highest explanatory proportion (51.52%) of TP. TPI displays a significant positive influence on TP, suggesting that environmental factors might indirectly influence community productivity through shaping biodiversity and species composition [63].

### 4.2. Altitudinal and Vegetation Effects on Forest Productivity

Our results demonstrated that there were differences in the ABP, BBP, and TP of woody plants at various altitude gradients, underscoring the important impacts of climate and environmental gradients associated with varying altitudes on forest productivity [64]. Specifically, our results are consistent with previous studies finding that ABP and TP tend to be the highest in low-altitude areas and lowest in high-altitude sites [65]. Warmer temperatures and low altitude probably provide favorable conditions for plants, which leads to increases in aboveground and total productivity [66]. Conversely, lower temperatures and precipitation in high-altitude areas might limit plant growth and productivity [67]. Mid-altitude areas might create more suitable environmental conditions and fewer disturbances, leading to relatively higher productivity. Productivity can also be influenced by the species composition and vegetation structure of various forest types. Mountain evergreen broad-leaved forests are richer in plant species than the other forest types, possibly contributing to higher biodiversity and productivity. In contrast, mountain evergreen dwarf forests, inhabiting higher altitudes and worse environmental conditions, might have fewer species and shorter vegetation, resulting in comparatively lower tree productivity.

The relationship between ABP, TP, and species richness is negative in low altitudes, which is potentially due to strong competition pressure or other ecological interactions [68]. This suggests that an increase in species richness in low-altitude areas could impose limitations on overall productivity due to intensified competition for resources [30]. Conversely, a higher species richness in mid- to high-altitude regions may have a beneficial impact on aboveground productivity, maybe as a result of complementarity effects or a greater diversity in resource utilization [61]. High species richness might enhance ecosystem stability, consequently enhancing total productivity [69].

Across different forest community types, our findings demonstrate that a negative correlation exists between productivity and species richness in VBF, while MBF, EMF, and DWF exhibit positive correlations between productivity and species richness. This could be related to the particular functional characteristics and species compositions of each of the forest types. Although VBFs might have a higher biodiversity, their relative poor productivity may be caused by issues including competition and human disturbances [70].

### 4.3. The Relative Influence of Productivity Explained by Different Groups of Predictors

The environmental factors had the highest importance for predicting total tree productivity. This could be due to the indirect influence of environmental factors on plant growth and resource utilization conditions impacting tree productivity [71]. TD contributes to 11.43% of the total impact on forest productivity. The influence of species diversity might involve factors such as species interactions and resource competition that impact overall tree productivity. The impact of community-weighted mean trait values on total tree productivity was the lowest. This suggests that the overall functional characteristics of the community have a minor influence on total tree productivity, and the productivity of the community as a whole could be more susceptible to the functional traits of dominant species [62]. PD exerted a similar impact on ABP and BBP. PD refers to the diversity of

species on an evolutionary tree, affecting species survival strategies and the expression of functional traits, and subsequently influencing the anatomic and physiological characteristics of the aboveground and belowground components of trees [72]. FD has the highest impact on ABP, constituting 18.73%. Functional diversity measures the variety of functional traits in the studied species, directly determining their resource utilization strategies [73]. According to this, interactions between trees in forest communities mostly take place in their aboveground parts, which promotes functional complementarity among them [74].

The structural equation model A reveals that all predictive variables, including environmental factors and biodiversity, collectively explain 22% of TP, 58% of ABP, and 52% of BBP. Conversely, the structural equation model B explains 22% of TP, 32% of ABP, and 71% of BBP. This underscores the significant explanatory power of these predictor variables, wherein environmental factors and biodiversity prominently influence aboveground and belowground productivity. Among the predictor variables, the group of taxonomic diversity indices demonstrate significant positive effects on both components of forest productivity. A community's likelihood of having high-productivity species is correlated with its biodiversity [75]. The production of the entire community is positively impacted by this, a phenomenon that can be explained by complementarity effects and selection mechanisms [61,62]. Conversely, the negative relationship observed between PD, FD, and productivity in these forests suggests that there is a considerable degree of species compositional heterogeneity, which in turn reflects a greater degree of divergence in the expression of evolutionary and functional traits among tree species. Productivity is negatively impacted when fewer species congregate around dominant ecological niches or functional groups [76]. Additionally, by altering the species composition, environmental factors may also have influence on community productivity. These findings suggest that the selection effects, complementarity effects, and the biomass-ratio hypothesis all affect tree productivity in the forest communities of the Nanling Mountains [61,62,77].

## 5. Conclusions

In this study, we investigated the factors that influence forest productivity in the subtropical forests of the Nanling Mountains, southern China. The environmental factors and a variety of diversity indices both play pivotal roles in explaining biomass productivity. The total productivity of forest trees is mostly influenced by environmental conditions; however, the effects of distinct forms of biodiversity on the different productivity components (aboveground and belowground) varied significantly. Forest production appears to be positively impacted by taxonomic biodiversity indices, whereas phylogenetic and functional diversity have the reverse effect. Moreover, these relationships seem also to vary depending on different altitudes. In low altitudes, negative effects of diversity on ABP and TP were found. Our primary findings in Nanling subtropical forests suggest that the excessive competition limits productivity, particularly in low-altitude regions. This implies that the existing diversity in low-altitude areas does not match the hypothesis of niche complementarity (i.e., it does not contribute to increased productivity). We claim that these results offer important new information for understanding the dynamics and functioning of southern China's forest ecosystems. However, there is still a significant degree of uncertainty surrounding these explanatory theories, especially in light of the inconsistent outcomes across research techniques, the choice of functional features, and the productivity estimation methodologies. Future research should further explore and validate the impacts of these factors on productivity, to gain a deeper comprehensive understanding of the intricacies of diversity–productivity relationships in forest ecosystems.

**Supplementary Materials:** The following supporting information can be downloaded at: https://www.mdpi.com/article/10.3390/f15030410/s1, Table S1 Basic location information of plots in the Nanling National Nature Reserve, southern China. Table S2 Allometric growth equation of different plants in Nanling. Table S3 Pearson correlation coefficients of environmental and biodiversity indices. Figure S1 Phylogenetic tree of the 201 species in this study.

**Author Contributions:** Conceptualization, W.X. and P.Z.; methodology, Z.L.; validation, P.Y.; investigation, Z.T.; writing—review and editing, W.X., P.Z. and M.Á.G.-R.; project administration, P.Z. All authors have read and agreed to the published version of the manuscript.

**Funding:** This study was supported by the Special Fund Project of Guangdong Academy of Sciences (2021GDASYL-20210103005), Funding Project for the Establishment of Comprehensive Industry Technology Innovation Center by Guangdong Academy of Sciences (2022GDASZH-2022010106), Science and Technology Project of Guangdong Province Natural Resources Department (GDZRZYKJ2023007), and the Special Project on National Science and Technology Basic Resources Investigation of China (2021FY100702).

**Data Availability Statement:** Data are available upon request.

**Acknowledgments:** We would like to thank the Guangdong Nanling Forest Ecosystem National Observation Research Station for support and the Nanling National Nature Reserve for field work assistance.

**Conflicts of Interest:** The authors declare no conflicts of interest.

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
