# Peer review of "Changes in Relationship between Forest Biomass Productivity and Biodiversity of Different Type Subtropical Forests in Southern China"

_forests, doi:10.3390/f15030410_

Round 1

Reviewer 1 Report

Comments and Suggestions for Authors

This work determines relationships between forest biomass production, biodiversity, and environmental factors in subtropical forests in southern China. Based on statistical analysis, the authors show a positive influence of taxonomy diversity on biomass production. The work seems to be innovative in the way it identifies factors affecting biomass productivity in forest ecosystems. The paper is well-written and contains all the designed sections. Also, the subject of the paper responds to the scope of the journal.

However, I have some questions and suggestions that can improve the quality of the work:

It’s not clear to me how phylogenetic analysis can add a positive interpretation to the result and output of this work. Please add some sentences in the introduction section to explain the utility of this parameter. In addition, how can you explain that taxonomic diversity has a positive effect on biomass production but that phylogenetic diversity has a negative effect? Also, I have some ambiguity about functional diversity.

Maybe you can explain more your opinion given in paragraph between L301 to L317 in the discussion section

A rapid check of your manuscript can also improve the quality of the paper.

In Fig.1 please add the meaning of the abbreviation

L124: don’t start a sentence with a number. you can add: total of 18715...

L238: Forstructural…For structural

In all figures, I prefer to see the meaning of the used abbreviations. These make easy the comprehension of the work.

Author Response

This work determines relationships between forest biomass production, biodiversity, and environmental factors in subtropical forests in southern China. Based on statistical analysis, the authors show a positive influence of taxonomy diversity on biomass production. The work seems to be innovative in the way it identifies factors affecting biomass productivity in forest ecosystems. The paper is well-written and contains all the designed sections. Also, the subject of the paper responds to the scope of the journal.

However, I have some questions and suggestions that can improve the quality of the work:

It’s not clear to me how phylogenetic analysis can add a positive interpretation to the result and output of this work. Please add some sentences in the introduction section to explain the utility of this parameter. In addition, how can you explain that taxonomic diversity has a positive effect on biomass production but that phylogenetic diversity has a negative effect? Also, I have some ambiguity about functional diversity.

Maybe you can explain more your opinion given in paragraph between L301 to L317 in the discussion section.

Response:Thank you for your advice. Although there was a positive effect on productivity of the middle subphylogenetic index, the overall phylogenetic diversity had a negative effect on productivity. The specific reason may be that the higher the structural phylogenetic diversity or functional diversity of the existing community species, the greater the possibility of functional redundancy. Therefore, we infered that there should be dominant functional groups in Nanling community, which dominate community productivity. As your suggestion,we have polished this part.

A rapid check of your manuscript can also improve the quality of the paper.

In Fig.1 please add the meaning of the abbreviation

Response:Thank you for your advice. As your suggestion,we have corrected this in L121-123.

L124: don’t start a sentence with a number. you can add: total of 18715...

Response:Thank you for your advice. As your suggestion,we have corrected this as “total of 18715...”

L238: Forstructural…For structural

 Response:Thank you for your advice. As your suggestion,we have corrected this mistake.

In all figures, I prefer to see the meaning of the used abbreviations. These make easy the comprehension of the work.

Response:Thank you for your advice. As your suggestion,we have added the meaning of the used abbreviations in figures.

Reviewer 2 Report

Comments and Suggestions for Authors

General comments:

This study addresses a relevant topic related to the relationships between biodiversity and forest productivity, a largely addressed topic in the forest ecology literature. An interesting approach followed by the author focuses on identifying the role of some drivers on forest productivity. The authors present a good introduction on the topic and the manuscript itself it is well structured and properly developed.

Although the research shows important findings, some methodological aspects needs more clarification to truly understand and assess the study results and conclusions. The following lines points out specific comments and suggestions to the authors.

Methods

Lines 106-108 Please indicates how many plots were included for each vegetation type, and indicate if such number of plots are sufficiently representative for each vegetation type in the study area.

Figure 1 does not show clearly the location of the 24 plots, perhaps a change in scale is required.

Lines 171-173, please indicate the number of plots allocated in each defined elevation range. Similar comments/observations to that for vegetation types. Providing such information would reinforce the study representativeness for each condition.

Although the list of functional traits in Table 1 seems exhaustive, some concerns arises on the procedure for estimating those traits for most species (155) for which specific information or studies does not exist, leading that values for those traits might be highly speculative. Perhaps a better approach would be focusing only on those species for which proper traits information is available, to avoid doubts on the certainty of the results from those traits and their true relationships with forest productivity. Interestingly the authors recognize the possible effect of this issue in their conclusions.

Results

Please indicate significant p-values in table 3 (TD and PD)

Figure 3, checking the box-plots location and data dispersion, it would be more valuable if the authors could identify which of these differences (in both, elevation range and vegetation type) are significant. A bit more of discussion on these particular results might provide more insights on true differences among those classes, thus reinforcing the results and conclusions.

Please check writing and spelling throughout the manuscript i.e. Lines 197 (might want to say “predictive” instead of “predicted”), Line 200, please check spelling (Table, instead of “Talbe”), Line 238.

Discussion

Although this section is overall well addressed, some additional comments or insights could enrich the analysis, considering that the study addressed the role of several factors or drivers to analyze trends in the biodiversity-productivity relationships.  i.e. based on the relevance of taxonomic diversity, how do the productivity relates with the other biodiversity metrics (Pielou evenness, Simpson´s dominance) with the suggested presence of high-productivity species in the community? (Lines 388-391). This kind of questions might provide more insights on the mechanisms suggested to understand the relationship between biodiversity and both above- and belowground productivity. 

Comments on the Quality of English Language

Minor editing and writing is required. See comments to the authors. 

Author Response

General comments:

This study addresses a relevant topic related to the relationships between biodiversity and forest productivity, a largely addressed topic in the forest ecology literature. An interesting approach followed by the author focuses on identifying the role of some drivers on forest productivity. The authors present a good introduction on the topic and the manuscript itself it is well structured and properly developed.

Although the research shows important findings, some methodological aspects needs more clarification to truly understand and assess the study results and conclusions. The following lines points out specific comments and suggestions to the authors.

Methods

Lines 106-108 Please indicates how many plots were included for each vegetation type, and indicate if such number of plots are sufficiently representative for each vegetation type in the study area.

Response:Thank you for your advice. As your suggestion,we have corrected this in lines 110-113.

Figure 1 does not show clearly the location of the 24 plots, perhaps a change in scale is required.

Response:Thank you for your advice. Beccausethe overlap of plots in the figure is difficult to solve by adjusting the scale, so we add the basic location information of the plots in the supplymentary Table S1.

Lines 171-173, please indicate the number of plots allocated in each defined elevation range. Similar comments/observations to that for vegetation types. Providing such information would reinforce the study representativeness for each condition.

Response:Thank you for your advice. As your suggestion,we have added the number of plots in each elevation range and the vegtation description of vegetation types in lines183-195 “The 7 plots in low-altitude gradient belongs to subtropical VBF(6) and MBF(1), which VBF can still keep the forest warm and humid in the dry season, and the vegetation shows some rainforest characteristics, such as the plate root phenomenon, and the trees are mainly composed of Lauraceae, Duyingaceae and Myrtaceae. The mid-altitude gradient has 11 plots, classified under MBF(7) and EMF(4). In Nanling Mountains, MBF   is mainly distributed in low hills below 1400m with occasional deciduous tree species, and most of the trees are Crustaceae. And EMF is mainly distributed on steep slopes at an altitude of 1000-1500m, and the common species are mainly Guangdong pine, Fujian cypress and long-bract hemlock. Within the high-altitude gradient, 6 plots are designated under DWF(3), EMF(2), and MBF(1). DWF is primarily distributed atop mountains exceeding 1500m, characterized by a perennially windy and cloudy climate. Trees in this region exhibit diminished stature, ranging from 4-8m, with rhododendrons constituting the predominant arboreal species.”

Although the list of functional traits in Table 1 seems exhaustive, some concerns arises on the procedure for estimating those traits for most species (155) for which specific information or studies does not exist, leading that values for those traits might be highly speculative. Perhaps a better approach would be focusing only on those species for which proper traits information is available, to avoid doubts on the certainty of the results from those traits and their true relationships with forest productivity. Interestingly the authors recognize the possible effect of this issue in their conclusions.

Response:Thank you for your advice.We acknowledge the concern raised about the exhaustive list of functional traits and the potential speculative nature of values for species lacking specific information.We are actively working to address this by intensifying our data collection efforts for the other 155 species in question.

Results

Please indicate significant p-values in table 3 (TD and PD)

Response:Thank you for your advice. As your suggestion,we have added the“*”to indicate the significant p-values.

Figure 3, checking the box-plots location and data dispersion, it would be more valuable if the authors could identify which of these differences (in both, elevation range and vegetation type) are significant. A bit more of discussion on these particular results might provide more insights on true differences among those classes, thus reinforcing the results and conclusions.

Response:Thank you for your advice. As your suggestion,we have added the significant indications in Figure 3.

Please check writing and spelling throughout the manuscript i.e. Lines 197 (might want to say “predictive” instead of “predicted”), Line 200, please check spelling (Table, instead of “Talbe”), Line 238.

Response:Thank you for your advice. As your suggestion, we have corrected these mistakes.

Discussion

Although this section is overall well addressed, some additional comments or insights could enrich the analysis, considering that the study addressed the role of several factors or drivers to analyze trends in the biodiversity-productivity relationships.  i.e. based on the relevance of taxonomic diversity, how do the productivity relates with the other biodiversity metrics (Pielou evenness, Simpson´s dominance) with the suggested presence of high-productivity species in the community? (Lines 388-391). This kind of questions might provide more insights on the mechanisms suggested to understand the relationship between biodiversity and both above- and belowground productivity. 

Response:Thank you for your advice. As your suggestion, we have polished this part in lines 425-428.

Reviewer 3 Report

Comments and Suggestions for Authors

The aim of the paper appears to be an in-depth exploration of the relationship between forest biodiversity and productivity, particularly in subtropical forests of southern China. Its strengths lie in its comprehensive review, consideration of multiple factors, and relevance to broader environmental issues.

Abstract: Be more specific in describing the background, dataset and methodology used. For example, mention the specific environmental factors and diversity metrics analyzed, providing readers with a clearer understanding of the research approach. Provide specific quantitative results where possible. Instead of stating that environmental factors have the greatest influence on total forest tree productivity, include relevant percentages or statistical measures to quantify this influence. I have found several abstracts that closely resemble yours in background  (e.g., https://doi.org/10.3390/f12080998 or https://doi.org/10.1111/1365-2745.13194 ). It is crucial to discern the scientific contribution your article aims to make.

Introduction: I have identified a notable weakness in the article, namely, the absence of a clear articulation of the overarching research purpose or the specific questions the authors aim to address. While the introductory section extensively discusses the relationship between forest biodiversity and productivity, the impact of environmental factors, and the influence of altitude gradients, it lacks a concise statement regarding the primary objectives of the study or the research questions guiding the investigation.

Materials and Methods: It is essential to enhance the clarity of the research methodology. For instance, the rationale behind mapping and measuring trees with a diameter of ≥1cm remains unclear (lines 114 – 115), as this data is neither aggregated nor analyzed in the results. Additionally, the biomass productivity calculations employed trees with a diameter at breast height (DBH) of ≥5cm (line 120). The origin and reliability of the data presented in Table 1 are not clearly delineated in the document.

The provided supplementary material lacks clarity in several aspects. The calculation of biomass for 201 species is unclear, particularly for conifers, where the leaf biomass equation appears to be incorrectly specified. Additionally, the presentation of the phylogenetic tree for the 201 species is illegible, making it challenging to comprehend the relationships among the species.

Results: The text is challenging to follow due to an extensive use of abbreviations, making it difficult to follow the narrative. Additionally, the described results in the text do not align with the information presented in the attached tables. For instance, the statement in lines 200-202 suggests that "CWM.rtn accounted for the majority of the variance... (coefficient=0.14, R2=0.31, p < 0.01)," but upon reviewing the tables, the reported numbers appear to be different. This inconsistency raises concerns about the accuracy of the reported results and hinders the reader's ability to reconcile the information presented in the text with the actual data in the tables. The figures and tables lack abbreviations, posing a significant challenge to comprehending the information without reading the entire manuscript. Figures 3 and 4 do not indicate biomass values on the y-axis.

It is unclear why the numbers of red circles between AGP and BBP sample plots in Figure 4 do not match, showing effect of altitude. It seems reasonable to expect that these numbers should match since they reflect sample plots. Additionally, the discrepancy in the trend of the Red TP line at low altitudes, following the BBP trend, raises questions, especially given that above-ground biomass typically constitutes the largest portion of the total biomass. These inconsistencies contribute to a lack of confidence in the presented results. Clarification and verification of the data and graphical representations are necessary to address these concerns and enhance the credibility of the findings.

Discussion: To strengthen the discussion section and tie it more explicitly to the results, consider incorporating references or specific connections to the relevant findings.

Conclusions: Consider refining certain phrases for precision. For instance, in the sentence "We consider that these findings furnish valuable insights," you might replace "consider" with a stronger verb like "assert" or "contend" for more assertiveness.

Overall, the article exhibits numerous grammatical errors that significantly impede comprehension. I strongly recommend thorough proofreading of the English language to rectify these issues before submitting the article. Some of the grammatical issues:

·        Incorrect spelling of words. E.g. Line 10 – diereminded

·        Too many or missing spaces between words. E.g. line 39 or 92

·        Abbreviations of words are not explained. E.g. line 59 “EFs”

·        Incorrect use of hyphens in text. E.g line 109 “veg-etation”, line 109 “ele-vation”

·        Incorrect use of italics. E.g. lines 284 and 285

·        Incorrect use of terminology. E.g lines 189 -193 Belowground biomass productivity is characterized by 3 different words “BBP, below ground, underground”

Comments on the Quality of English Language

·        Incorrect spelling of words. E.g. Line 10 – diereminded

·        Too many or missing spaces between words. E.g. line 39 or 92

·        Abbreviations of words are not explained. E.g. line 59 “EFs”

·        Incorrect use of hyphens in text. E.g line 109 “veg-etation”, line 109 “ele-vation”

·        Incorrect use of italics. E.g. lines 284 and 285

·        Incorrect use of terminology. E.g lines 189 -193 Belowground biomass productivity is characterized by 3 different words “BBP, below ground, underground”

Author Response

The aim of the paper appears to be an in-depth exploration of the relationship between forest biodiversity and productivity, particularly in subtropical forests of southern China. Its strengths lie in its comprehensive review, consideration of multiple factors, and relevance to broader environmental issues.

Abstract: Be more specific in describing the background, dataset and methodology used. For example, mention the specific environmental factors and diversity metrics analyzed, providing readers with a clearer understanding of the research approach. Provide specific quantitative results where possible. Instead of stating that environmental factors have the greatest influence on total forest tree productivity, include relevant percentages or statistical measures to quantify this influence. I have found several abstracts that closely resemble yours in background  (e.g., https://doi.org/10.3390/f12080998 or https://doi.org/10.1111/1365-2745.13194 ). It is crucial to discern the scientific contribution your article aims to make.

Response:Thank you for your advice. As your suggestion,we have corrected the abstract part.

Introduction: I have identified a notable weakness in the article, namely, the absence of a clear articulation of the overarching research purpose or the specific questions the authors aim to address. While the introductory section extensively discusses the relationship between forest biodiversity and productivity, the impact of environmental factors, and the influence of altitude gradients, it lacks a concise statement regarding the primary objectives of the study or the research questions guiding the investigation.

Response:Thank you for your advice. As your suggestion,we have added a paragraph to explain the research purpose and questions.

Materials and Methods: It is essential to enhance the clarity of the research methodology. For instance, the rationale behind mapping and measuring trees with a diameter of ≥1cm remains unclear (lines 114 – 115), as this data is neither aggregated nor analyzed in the results. Additionally, the biomass productivity calculations employed trees with a diameter at breast height (DBH) of ≥5cm (line 120). The origin and reliability of the data presented in Table 1 are not clearly delineated in the document.

Response:Thank you for your advice. The measurement of tree diameter at breast height (DBH) was conducted using standard circumference tapes with the initial DBH for each tree measured starting at 1 cm. During data analysis, we observed limited growth data for trees with a DBH below 5 cm in both periods. Consequently, we opted to focus our study on trees with a DBH of 5 cm and above, considering them as the effective subjects for the analysis. And the data in Table 1 comes from our own samples measurement, data of unmeasured species from genera means of meansured species.

The provided supplementary material lacks clarity in several aspects. The calculation of biomass for 201 species is unclear, particularly for conifers, where the leaf biomass equation appears to be incorrectly specified. Additionally, the presentation of the phylogenetic tree for the 201 species is illegible, making it challenging to comprehend the relationships among the species.

Response: Thank you for your advice. The biomass equation used in this study is mainly derived from the book “Carbon reserves of forest ecosystem in China: biomass equation” and other literatures. Because of the law and the reality, we don't actually cut down trees to build biomass calculation formulas. We will continue to collect relevant data to solve this problem.

Results: The text is challenging to follow due to an extensive use of abbreviations, making it difficult to follow the narrative. Additionally, the described results in the text do not align with the information presented in the attached tables. For instance, the statement in lines 200-202 suggests that "CWM.rtn accounted for the majority of the variance... (coefficient=0.14, R2=0.31, p < 0.01)," but upon reviewing the tables, the reported numbers appear to be different. This inconsistency raises concerns about the accuracy of the reported results and hinders the reader's ability to reconcile the information presented in the text with the actual data in the tables. The figures and tables lack abbreviations, posing a significant challenge to comprehending the information without reading the entire manuscript. Figures 3 and 4 do not indicate biomass values on the y-axis.

Response:Thank you for your advice. As your suggestion,we have corrected these copy errors, added abbreviations and corrected the y-axis of Figures 3 and 4.

It is unclear why the numbers of red circles between AGP and BBP sample plots in Figure 4 do not match, showing effect of altitude. It seems reasonable to expect that these numbers should match since they reflect sample plots. Additionally, the discrepancy in the trend of the Red TP line at low altitudes, following the BBP trend, raises questions, especially given that above-ground biomass typically constitutes the largest portion of the total biomass. These inconsistencies contribute to a lack of confidence in the presented results. Clarification and verification of the data and graphical representations are necessary to address these concerns and enhance the credibility of the findings.

Response:Thank you for your advice. Based on your suggestion, we checked the code of Figure 4 and it is true that there is an error in the TP diagram, and we have replaced the correct figure4. As your suggestion,we also have added the units of y-axis.

Discussion: To strengthen the discussion section and tie it more explicitly to the results, consider incorporating references or specific connections to the relevant findings.

Response:Thank you for your advice. As your suggestion,we have polished the discussion part.

Conclusions: Consider refining certain phrases for precision. For instance, in the sentence "We consider that these findings furnish valuable insights," you might replace "consider" with a stronger verb like "assert" or "contend" for more assertiveness.

Response:Thank you for your advice. As your suggestion,we have corrected it.

Overall, the article exhibits numerous grammatical errors that significantly impede comprehension. I strongly recommend thorough proofreading of the English language to rectify these issues before submitting the article. Some of the grammatical issues:

  • Incorrect spelling of words. E.g. Line 10 – diereminded

Response:Thank you for your advice. As your suggestion, we have corrected it.

  • Too many or missing spaces between words. E.g. line 39 or 92

Response:Thank you for your advice. As your suggestion, we have corrected it.

  • Abbreviations of words are not explained. E.g. line 59 “EFs”

Response:Thank you for your advice. As your suggestion, we have corrected it.

  • Incorrect use of hyphens in text. E.g line 109 “veg-etation”, line 109 “ele-vation”

Response:Thank you for your advice. As your suggestion, we have corrected it.

  • Incorrect use of italics. E.g. lines 284 and 285

Response:Thank you for your advice. As your suggestion, we have corrected it.

  • Incorrect use of terminology. E.g lines 189 -193 Belowground biomass productivity is characterized by 3 different words “BBP, below ground, underground”

Response:Thank you for your advice. As your suggestion, we have corrected it.

Round 2

Reviewer 3 Report

Comments and Suggestions for Authors

I would like to reiterate the main concerns contributing to my recommendation:

    Insufficient Addressing of Objections:

    The authors have not adequately addressed the objections raised during the initial review. The corrections made seem superficial, overlooking significant flaws in the article.

    Concerns Regarding Data Integrity:

    I still harbor reservations about the accuracy and integrity of the data presented in the tables and figures. There seems to be a lack of transparency, and I suspect potential manipulation of the data.

    Persistence of Grammatical and Other Errors:

    The manuscript continues to exhibit numerous grammatical and other errors, which have not been sufficiently rectified. These errors impede the clarity and professionalism of the manuscript.

    Improper use of Terminology:

    Throughout the manuscript, the authors consistently misuse the terms "Biomass production" and "biomass productivity." It is evident that they lack a clear understanding of these related concepts, leading to confusion in the context of biomass.

Given these persistent issues, I cannot endorse the publication of the article in its current form. I recommend that the authors address these concerns comprehensively before considering it for publication. Further scrutiny and revisions are essential to uphold the standards of accuracy, clarity, and professionalism expected in scientific publications.

Thank you for your understanding.

Comments on the Quality of English Language

Persistence of Grammatical and Other Errors:

    The manuscript continues to exhibit numerous grammatical and other errors, which have not been sufficiently rectified. These errors impede the clarity and professionalism of the manuscript.

Author Response

I would like to reiterate the main concerns contributing to my recommendation:

    Insufficient Addressing of Objections:

    The authors have not adequately addressed the objections raised during the initial review. The corrections made seem superficial, overlooking significant flaws in the article.

     Concerns Regarding Data Integrity:

I still harbor reservations about the accuracy and integrity of the data presented in the tables and figures. There seems to be a lack of transparency, and I suspect potential manipulation of the data.

Response: Thank you for your advice. We can guarantee that the data and results of this paper did not have data manipulation and falsification. If necessary, we can agree to submit the original data. We have made further changes to the pictures and tables and the description of the results that caused the doubt in the hope that the results will be satisfactory.

     Persistence of Grammatical and Other Errors:

The manuscript continues to exhibit numerous grammatical and other errors, which have not been sufficiently rectified. These errors impede the clarity and professionalism of the manuscript.

Response: Thank you for your advice. According to your suggestion, we have invited experts to improve the English grammar and writing quality of the manuscript.

     Improper use of Terminology:

Throughout the manuscript, the authors consistently misuse the terms "Biomass production" and "biomass productivity." It is evident that they lack a clear understanding of these related concepts, leading to confusion in the context of biomass.

Response: Thank you for your advice. According to your suggestion, we have corrected all "production"to "productivity" in the manuscript.

Given these persistent issues, I cannot endorse the publication of the article in its current form. I recommend that the authors address these concerns comprehensively before considering it for publication. Further scrutiny and revisions are essential to uphold the standards of accuracy, clarity, and professionalism expected in scientific publications.

Response: Thank you for your advice. According to your suggestion, we have revised the whole manuscript and invited experts to improve the grammar and quality of the manuscript. At the same time, in order to increase the accuracy and reliability of the data, the appendix table of the relevant parameters of the variables is added to the appendix material. We can guarantee that there is no data manipulation in the process of data calculation. We are also continuing to collect data on species diversity and functional traits in the Nanling Mountains for future research.

Thank you for your understanding.
